# 30-Min Exposure to Tobacco Smoke Influences Airway Ion Transport—An In Vitro Study

Katarzyna Henke [1], Irena Balcerzak [1], Ewa Czepil [1], Alicja Bem [1], Elżbieta Piskorska [1], Dorota Olszewska-Słonina [1], Alina Woźniak [2], Karolina Szewczyk-Golec [2] and Iga Hołyńska-Iwan [1,*]

[1]  Department of Pathobiochemistry and Clinical Chemistry, Faculty of Pharmacy, Ludwik Rydygier Collegium Medicum in Bydgoszcz, Nicolaus Copernicus University in Toruń, 85-094 Bydgoszcz, Poland; 304775@stud.umk.pl (K.H.); 304754@stud.umk.pl (I.B.); 304760@stud.umk.pl (E.C.); 304757@stud.umk.pl (A.B.); piskorska_e@cm.umk.pl (E.P.); dorolsze@cm.umk.pl (D.O.-S.)

[2]  Department of Medical Biology and Biochemistry, Faculty of Medicine, Collegium Medicum in Bydgoszcz, Nicolaus Copernicus University in Toruń, 85-092 Bydgoszcz, Poland; alina-wozniak@wp.pl (A.W.); karosz@cm.umk.pl (K.S.-G.)

\*  Correspondence: igaholynska@cm.umk.pl

**Abstract: Introduction:** Smoking is one of the most important causes of cancer in humans. However, it has not been proven how long exposure to cigarette smoke is sufficient to induce cancerogenesis. Cigarette smoke can cause changes in ion and water transport and the maintenance of mucociliary transport. The conducted research concerned the assessment of changes in ion transport in rabbit tracheal specimens after 30 min of exposure to cigarette smoke. **Materials and Methods:** A modified Ussing chamber was used to measure the transepithelial electrical potential under stationary conditions (PD) and during mechanical stimulation (PDmin), and the transepithelial electrical resistance (R) in control and cigarette smoke-exposed tracheal fragments. **Results:** Significant changes in PD ($-2.53$ vs. $-3.92$ mV) and PDmin ($-2.74$ vs. $-0.39$ mV) were noted for the samples exposed to smoke, which can be associated with a rise in reactivity after applying a mechanical stimulus. In addition, the measured R (108 vs. 136 $\Omega/cm^2$) indicated no changes in the vitality of the samples, but an increase in their permeability to ions in the experimental conditions. **Conclusions:** A single 30-min exposure to cigarette smoke has been shown to be associated with increased permeability of the tracheal epithelium to ions and thus to substances emitted during smoking, which might be sufficient to create the possibility of initiating procarcinogenic processes.

**Keywords:** airways; CFTR; chloride; ENaC; ion transport; smoking; sodium; transepithelial potential; transepithelial resistance

## 1. Introduction

Epidemiological studies have proven the adverse health effects of cigarette smoking for decades [1–3]. According to World Health Organization (WHO), there are nearly 6 million smoking-related deaths per year and this number is expected to rise to about 8 million by the end of 2030 [2].

Smoking is a major risk factor for many civilization diseases such as cancer, lung diseases, including chronic obstructive pulmonary disease (COPD), and cardiovascular diseases, for which smoking is the main cause [1–5]. Experimental and epidemiological studies have shown that exposure to environmental tobacco smoke (ETS), commonly referred to as second-hand smoke (SHS) or passive smoking, also causes respiratory and cardiac diseases, including lung cancer in adult non-smokers [1,2,4,5]. In children, SHS impairs lung development, promotes the development of hypersensitivity and/or allergies, and asthma, as well as increases the risk of sudden infant death syndrome [2,6]. Numerous studies have proven that smoking not only affects physical health but also leads to neurological impairment, including memory and attention deficits [2]. Smoking contributes

to approximately 30% of all cancer-related deaths in the developed countries. Inhaling the fumes causes lung carcinoma in both women and men, which is the dominant tumor among malignancies [1,5]. The usage of cigarettes entails the risk of oral cancer, including the lips and tongue; it also increases the risk of sinus and nasal diseases. Traditional cigarettes are responsible for the cancer of the nasopharynx and larynx and contribute to the development of diseases of the mouth and throat. Smoking causes esophageal malignant tumors, especially squamous cell carcinoma, and contributes to the formation of esophageal adenocarcinoma [1,3,5]. Similarly, pancreatic and stomach cancer can start with cigarette consumption.

Cigarette smoke contains more than 60 well-known carcinogens [1,2,4]. The term carcinogen means any chemical, physical, or viral agent that starts cancer or increases its incidence. Chemical carcinogens found in cigarette smoke contribute to the onset of cancer [1,2,4]. Cigarette fumes cause increased production of substances that trigger a complex inflammatory response, leading to the structural changes in the airways [2,4]. Tobacco smoke causes irritation and damage to the cells of the respiratory epithelium, hypertrophy of glandular cells, and an increase in their number in the bronchial tree, as well as damage to the ciliated epithelium. Inflammatory peritoneal lesions leading to fibrosis also can occur. Mentioned structural changes found in smokers may be caused by various components present in cigarette smoke such as acetic aldehyde, formaldehyde, acrolein, and free radicals [7].

Cigarette smoke is a complex mixture of many toxic and carcinogenic substances, containing about 8000 chemicals produced during the combustion of tobacco both during the oxidation and the inhalation of tobacco smoke (Table 1) [5,6]. Chemicals are divided into mainstream smoke (MS), sidestream smoke (SS), secondhand smoke (SHS), thirdhand smoke (THS), and discarded cigarette butts (CBS) [3]. Traditional cigarettes are a mixture of dried and cured tobacco leaves, which are rolled up into a thin smoking paper [2,3,5]. Traditional tobacco cigarettes burn at around 800 °C, producing nicotine-containing smoke and harmful chemicals [6].

The transport of ions in the respiratory epithelium occurs through specific mechanisms, which are necessary for the secretion of chlorides, the absorption of sodium, and the extracellular transport of potassium ions [8–14]. Because of these mechanisms, it is possible to regulate the amount, viscosity, and composition of airway surface liquid (ASL). Respiratory fluid plays a significant role in defense mechanisms of the lungs [8,12–16]. Mucosal ciliary cleansing is crucial to protect the respiratory tract from infection and the harmful effects of smoke and various inhaled irritants that cause augmented oxidative stress and chronic inflammation [16–18]. Oxidative stress has both acute and long-term effects on airway ion transport functions [16]. The absorption of sodium ions consists of two steps and is conditioned by the action of the epithelial sodium channel (ENaC) on the apex side and the sodium-potassium pump on the basal-lateral side of the ciliary epithelial cells [10,12]. The sodium channel is permeable to $H^+$, $Li^+$, and $Na^+$ and impermeable to $K^+$, and $NH_4^+$ due to their large size [12,19]. Tumor transformation is associated with the changes in sodium and potassium transport from blood and through intracellular membranes [9,20]. These changes are crucial to the cancer reprogramming of cells that contribute to cancer initiation, progression, tumor proliferation, metastasis, and resistance to anticancer therapy [20,21]. Changing the activity of selected channels and the transport of those ions to and from respiratory epithelial cells can be used to develop new therapeutic strategies, for example by re-targeting already existing drugs that are known to target potassium channels [21,22].

Cigarette smoke causes many qualitative changes in the ASL mechanism, which in turn contributes to the better penetration of substances from the smoke [22]. The changes include damage to the epithelium, overproduction of mucins and mucus, changes in the concentration of calcium, and an increase in bacterial colonization [23]. The disturbances in the normal flow of ions lead to changes in ASL, and consequently to the development of an inflammatory response and cough initiation. Increased cough reflex combined with

changes in ASL makes coughing more difficult, which is observed in conditions such as asthma, cystic fibrosis, and COPD [18,24]. Accordingly, cigarette smoke has an adverse influence on the respiratory tract, as confirmed by numerous studies [1,5]. The effect of cigarette smoke on the ion transport in the airway epithelium has not been thoroughly studied. Therefore, the aim of the presented study was to assess the effect of 30-min exposure to cigarette smoke on the transport of ions in the respiratory tract.

**Table 1.** Selected carcinogenic compounds in cigarette smoke from non-filter cigarettes [1,4,5].

| Chemical Compound | IARC [1] Group | Chemical Compound | IARC Group |
|---|---|---|---|
| 4-Aminobiphenyl | 1 [2] | Dibenzo[a,j]acridine | 2B |
| 2-Naphthylamine | 1 | Dibenzo[c,g]carbazole | 2B |
| Benzene | 1 | Furan | 2B |
| Chloroethylene | 1 | N-Nitrosoethylmethylamine | 2B |
| Ethylene oxide | 1 | N-Nitroso-di-n-butylamine | 2B |
| Arsenic | 1 | N-Nitrosopyrrolidine | 2B |
| Beryllium | 1 | N-Nitrozonornicotine | 2B |
| Nickel | 1 | 2-Toluidine | 2B |
| Chromium(VI) | 1 | 2,6-Dimethylaniline | 2B |
| Cadmium | 1 | Acetaldehyde | 2B |
| Polonium-210 | 1 | 1,3-Butadiene | 2B |
| Benz[a]anthracene | 2A [3] | Isoprene | 2B |
| Benzo[a]pyrene | 2A | Styrene | 2B |
| Dibenz[a,h]anthracene | 2A | Acetamide | 2B |
| N-Nitrosodimethylamine | 2A | DDT | 2B |
| N-Nitrosodiethylamine | 2A | DDE | 2B |
| Formaldehyde | 2A | Pyrocatechol | 2B |
| Acrylonitrile | 2A | Nitromethane | 2B |
| Benzo[b]fluoranthene | 2B [4] | 2-Nitropropane | 2B |
| Benzo[j]fluoranthene | 2B | Nitrobenzene | 2B |
| Benzo[k]fluoranthene | 2B | Ethyl carbamate | 2B |
| Dibenzo[a,l]pyrene | 2B | Propylene oxide | 2B |
| Indeno [1,2,3-cd]pyrene | 2B | Hydrazine | 2B |
| 5-Methylchrysene | 2B | Cobalt | 2B |
| Dibenzo[a,h]acridine | 2B | Lead | 2B |

[1] IARC—the International Agency for Research on Cancer, [2] group 1—cancerogenic, [3] group 2A—probably cancerogenic, [4] group 2B—possibly cancerogenic.

## 2. Materials and Methods

The experiment was performed on 108 tracheal fragments of 36 New Zealand albino rabbits. The study subjects included adult rabbits of both sexes, weighing 3.5 to 4.0 kg, aged three to four months. The animals were asphyxiated with a high concentration of $CO_2$, with the gas concentration increasing to 60% of the inhaled air. The death of the animals was confirmed by a qualified person. The tracheae were excised and immediately placed in the Ringer's solution (RS), then cut along the membranous part and divided into 3 fragments. It has been previously shown that specimens prepared in this way contain intact epithelium and nerve fibers [18,24]. The fragments were transferred to the incubation solutions according to the experimental schedule and then mounted in a modified Ussing apparatus.

### 2.1. Chemicals and Solutions

The following chemicals and solutions were used in the experiment:

(1) RS—Ringer's solution—$K^+$ 4.0 mM; $Na^+$ 147.2 mM; $Ca^{2+}$ 2.2 mM; $Mg^{2+}$ 2.6 mM; $Cl^-$ 160.8 mM; HEPES (4-(2-hydroxyethyl)piperazine-1-ethanosulfonic acid; 10.0 mM; Sigma-Aldrich, Burlington, Massachusetts, USA), adjusted to pH 7.4; a basic solution with iso-osmotic properties. Mineral compounds (KCl, NaCl, $CaCl_2$, $MgCl_2$) were purchased from Avantor Performance Materials Poland.

(2)     A—amiloride solution; 266.09 g/mol (Sigma-Aldrich, Burlington, MA, USA); used as an inhibitor of transepithelial transport of sodium ions, dissolved and diluted in RS (0.1 mmol/L).

(3)     B—bumetanide solution; 364.42 g/mol (Sigma-Aldrich, Burlington, MA, USA); used as an inhibitor of transepithelial transport of chloride ions dissolved in DMSO and diluted in RS (0.1 mmol/L).

*2.2. Experimental Procedure*

The cleaned tracheal specimens randomly selected for the experimental groups were immersed in RS and placed in a restricted airflow chamber and exposed to smoke from two cigarettes, mimicking SHS. The trachea samples were positioned 10 cm above the smoldering cigarettes. After 30 min of exposure to smoke, the samples were mounted in a fluid-filled chamber according to the experimental procedure.

The experiment used a modified Ussing chamber, equipped with a nozzle connected to a peristaltic pump. Due to the use of a modified Ussing chamber, it is possible to measure changes in the transepithelial electric potential, reflecting the transport of ions, under the influence of the tested substances [9,14,18,24]. The modified Ussing chamber is made of two symmetrical chambers, which enables electrical isolation and incubation of horizontally arranged tissue samples in fluid, in this case, a fragment of the trachea. In addition, the modification enabled the use of a mechanical-chemical stimulus, i.e., gentle washing of the mucous side of the sample with fluid from the peristaltic pump, in accordance with the experimental procedure [9,14,18,24].

Tissue samples were mounted horizontally. The nozzle was placed at a distance of 3–5 mm from the mucous surface of the trachea, and the stream of stimulating fluid from the peristaltic pump was directed in the direction perpendicular to the tissue section, gently washing the specimens. The mechanical-chemical stimulus was applied to the mucous side of the tissue, which ensures the effects of solutions and smoke on receptors and ion channels. Stimulation was carried out with appropriate solutions, fluid flowing from the nozzle in a volume of 0.06 mL/s (1 mL/15 s) at 25 °C. Each stimulation lasted 15 s, which was sufficient to induce reproducible and measurable changes in the transepithelial electric potential difference (PD). The control group was treated in the same way, except for 30 min of exposure to cigarette smoke.

Ag/AgCl electrodes and agar bridges ensured the connection of the modified Ussing chamber with the measuring apparatus. Electrophysiological parameters were assessed continuously, the examination for each fragment lasted 20 min. The measurement of PD (mV) at stationary conditions reflects the constant transport of ions in the sample after the application of the incubation fluid. The lowest transepithelial potential (PDmin, mV) was measured during 15 s of stimulation of the tracheal fragments with the solution provided in the experimental plan. The electrical resistance of the sample (R, $\Omega/cm^2$) was determined by applying a current of 10 μA to the tissue stimulus and measuring the corresponding change in voltage according to Ohm's law. The experimental procedure for a single tissue fragment lasted approximately 60 min at 25 °C and included incubation and chamber testing (Figure 1).

*2.3. Data Analysis*

Data were recorded in the experimental protocol EVC 4000 (WPI, Worcester, Massachusetts, USA), which was connected to the MP150 data acquisition system and transferred to the AcqKnowledge 3.8.1 computer software (Biopac Systems, Inc., Goleta, California, USA). Statistical analysis was performed in Statistica 13.1 (StatSoft, Inc., Kraków, Poland). The non-parametrical data distribution was confirmed by the Kolmogorov-Smirnov test, with the Lilliefors correction. The Wilcoxon test was used to compare data from the same incubation conditions in the control and smoke-treated groups, with a significance of $p < 0.05$. The Mann-Whitney test was used to detect significant differences (at $p < 0.05$) for different experimental conditions.

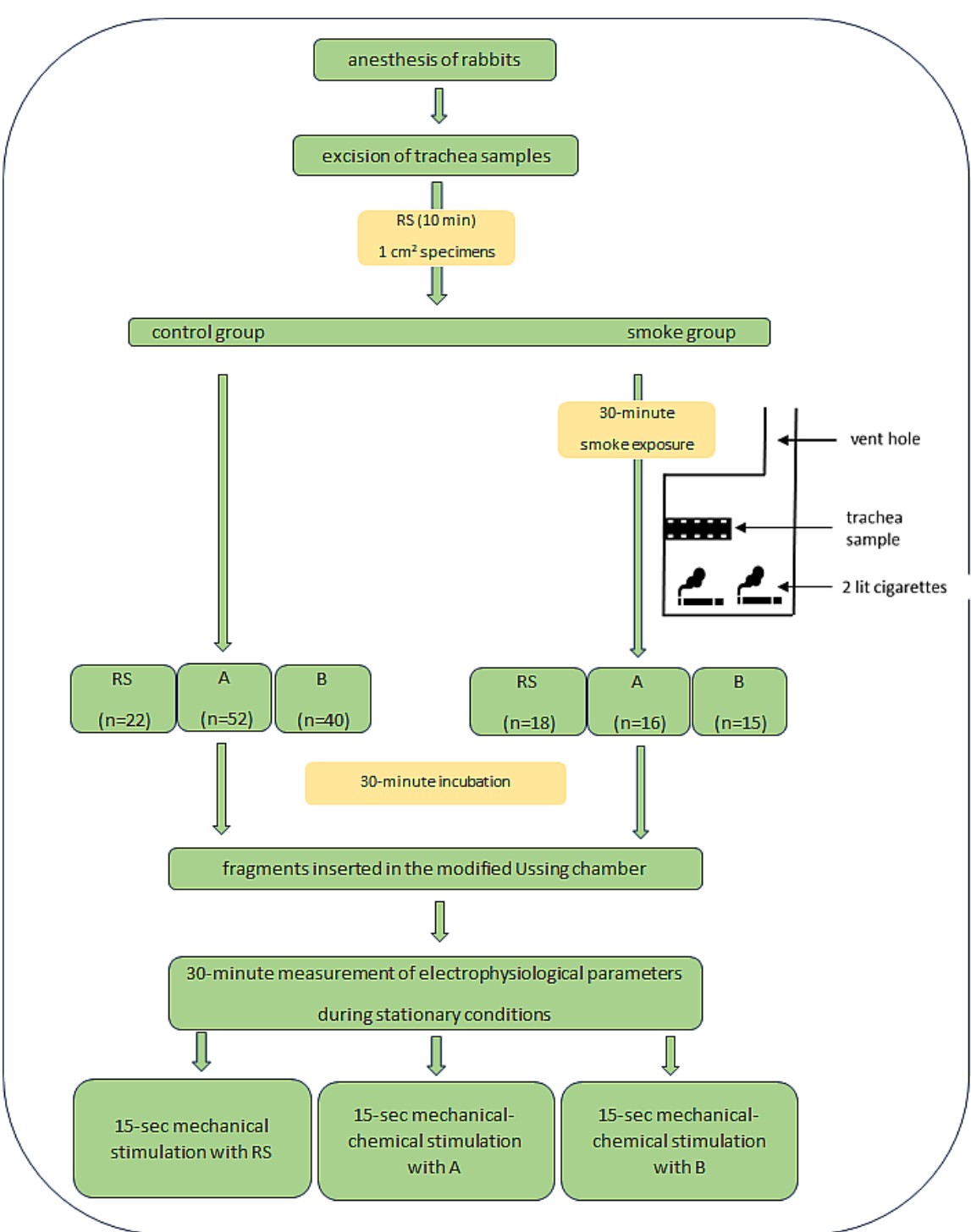

**Figure 1.** Study design. Abbreviation: RS—Ringer's solution, A—amiloride solution (0.1 mmol/L), B—bumetanide solution (0.1 mmol/L).

## 3. Results

The R measurement of smoke-treated and untreated tissue samples ranged from 79 $\Omega/cm^2$ of tissue exposed to smoke in the incubation B to 239 $\Omega/cm^2$ for tissues that were not treated with cigarette smoke and incubated in RS (see Table 2). The R measurements were possible and showed no damage or deformation for all examined tissue fragments. The smoke-treated samples showed constant R values throughout the experiment (Wilcoxon test). However, the resistance of tissues that were not exposed to cigarette smoke was

significantly lower for the incubation in RS and chloride ion transport inhibitor (the B solution) compared to the control. No significant differences were observed for the samples exposed to cigarette smoke and a solution of amiloride, a sodium ion transport inhibitor (Mann-Whitney test).

**Table 2.** Values of transepithelial resistance (R, $\Omega/cm^2$) and transepithelial electric potential (PD, mV) were measured in stationary conditions of the tracheal fragments unexposed and exposed to cigarette smoke for 30 min.

| | | | | Stationary Conditions | | | | | | | |
|---|---|---|---|---|---|---|---|---|---|---|---|
| | **Incubation** | | | **Ctr** | | **Smoke** | | **Ctr** | | **Smoke** | |
| | **Ctr (n)** | **Smoke (n)** | | **R Initial** | **R Final** | **R Initial** | **R Final** | **PD Initial** | **PD Final** | **PD Initial** | **PD Final** |
| **RS** | 22 | 18 | median | 185 | 136 | 110 | 108 | −3.92 | −1.48 | −2.53 | −2.32 |
| | | | upper quartile | 239 | 152 | 152 | 151 | −2.43 | −0.7 | −1.08 | −0.96 |
| | | | lower quartile | 144 | 108 | 91 | 88 | −5.10 | −1.74 | −3.61 | −4.23 |
| **A** | 52 | 16 | median | 135 | 123 | 139 | 128 | −3.41 | −2.15 | −2.07 | −1.89 |
| | | | upper quartile | 153 | 139 | 192 | 164 | −2.72 | −1.4 | −1.5 | −1.37 |
| | | | lower quartile | 102 | 110 | 108 | 110 | −4.50 | −2.63 | −3.24 | −2.76 |
| **B** | 40 | 15 | median | 147 | 155 | 79 | 80 | −1.77 | −1.63 | −0.64 | −0.53 |
| | | | upper quartile | 168 | 169 | 87 | 87 | −1.19 | −1.37 | −0.32 | −0.26 |
| | | | lower quartile | 122 | 135 | 73 | 73 | −2.60 | −2.52 | −1.42 | −1.34 |

Abbreviations: RS—iso-osmotic Ringer solution, A—amiloride solution (0.1 mmol/L), B—bumetanide solution (0.1 mmol/L), Ctr—control conditions (trachea samples incubated in RS, A and B solution), Smoke—smoking conditions (trachea samples exposed to cigarette smoke and incubated in RS, A and B solution).

The initial and final PD showed no statistically significant differences for incubation in RS and in the conditions of inhibition of the chloride ion transport path (B solution) for both the controls and the cigarette smoke-treated samples. Conversely, for both unexposed and exposed to cigarette smoke trachea samples, incubated in the A solution, an increase in PD initial and final was observed (Wilcoxon test). The lowest PD of −5.10 mV was recorded for the control tissues held in RS, and the lowest PD of −3.61 mV was also measured in RS for the samples treated with smoke. The highest PD value −0.26 mV was obtained for tissues treated with cigarette smoke and incubated in solution B, under the conditions of inhibited transepithelial transport of chloride ions. Comparing the PD of smoke-treated and control tracheas, a significant increase in potential was observed for all incubation conditions (Mann-Whitney test).

For all experimental conditions, the applied 15-s stimulus, imitating the gentle flow of fluid on the mucous surface of tracheal specimens, caused repetitive tissue reactions related to the change in ion transport occurring in the examined tracheal fragments (Table 3). Both for the unexposed tracheal fragments and those exposed to cigarette smoke for 30 min, the PDmin measured during stimulation was different from PD measured in stationary conditions i.e., without stimulation (Table 4, Wilcoxon test). The responses of tissues exposed to cigarette smoke were significantly higher during incubation and stimulation with iso-osmotic RS. On the other hand, the use of sodium (A solution) and chloride (B solution) ion transport inhibitors resulted in a significant increase in the measured PDmin (Table 5, Mann-Whitney test). In the presence of the ion transport inhibitors, the mechanisms regulating ion transport to maintain the negative surface charge of the cells were weakened, as evidenced by significant differences in the measured potential, and exposure to cigarette smoke exacerbated this response.

**Table 3.** Values of the minimum transepithelial electric potential (PDmin, mV) measured during 15 s mechanical and/or mechanical-chemical stimulation of tracheal specimens unexposed and exposed to cigarette smoke for 30 min.

| | | | | Stimulation Conditions | |
|---|---|---|---|---|---|
| | | | | Ctr | Smoke |
| Incubation | Ctr (n) | Smoke (n) | | PDmin | PDmin |
| **RS** | 22 | 18 | median | −0.39 | −2.74 |
| | | | upper quartile | −0.21 | −1.28 |
| | | | lower quartile | −0.61 | −4.73 |
| **A** | 52 | 16 | median | −3.41 | −2.17 |
| | | | upper quartile | −2.72 | −1.47 |
| | | | lower quartile | −4.85 | −3.36 |
| **B** | 40 | 15 | median | −1.8 | −0.95 |
| | | | upper quartile | −1.43 | −0.37 |
| | | | lower quartile | −2.83 | −1.53 |

Abbreviations: RS—iso-osmotic Ringer solution, A—amiloride solution (0.1 mmol/L), B—bumetanide solution (0.1 mmol/L), Ctr—control conditions (trachea samples incubated in RS, A and B solution), Smoke—smoking conditions (trachea samples exposed to cigarette smoke and incubated in RS, A and B solution).

**Table 4.** Results of the Wilcoxon test of transepithelial resistance (R, $\Omega/\text{cm}^2$) and transepithelial electric potential (PD, mV) measured in stationary conditions and the minimal transepithelial electric potential (PDmin, mV) measured during 15-s mechanical and/or mechanical-chemical stimulation for tracheal fragments unexposed and exposed to cigarette smoke for 30 min.

| | Control | | | Smoke | | |
|---|---|---|---|---|---|---|
| | R Initial vs. R Final | PD Initial vs. PD Final | PD vs. PDmin | R Initial vs. R Final | PD Initial vs. PD Final | PD vs. PDmin |
| **RS** | 0.0096 | 0.0731 | <0.001 | 0.1240 | 0.0535 | <0.001 |
| **A** | 0.4029 | <0.001 | <0.001 | 0.8445 | <0.001 | <0.001 |
| **B** | 0.0788 | 0.2304 | 0.0018 | 0.2209 | 0.1094 | <0.001 |

Abbreviations: RS—iso-osmotic Ringer solution, A—amiloride solution (0.1 mmol/L), B—bumetanide solution (0.1 mmol/L), Ctr—control conditions (trachea samples incubated in RS, A and B solution); Smoke—smoking conditions (trachea samples exposed to cigarette smoke and incubated in RS, A and B solution).

**Table 5.** Results of the Mann-Whitney test of transepithelial resistance (R, $\Omega/\text{cm}^2$) and transepithelial electric potential (PD, mV) measured in stationary conditions and the minimal transepithelial electric potential (PDmin, mV) measured during 15-s mechanical and/or mechanical-chemical stimulation for tracheal fragments unexposed and exposed to cigarette smoke for 30 min.

| | Control vs. Smoke | | |
|---|---|---|---|
| | RS | A | B |
| **R** | <0.001 | 0.9905 | <0.001 |
| **PD** | 0.0278 | <0.001 | <0.001 |
| **PDmin** | <0.001 | <0.001 | <0.001 |

Abbreviations: RS—iso-osmotic Ringer solution, A—amiloride solution (0.1 mmol/L), B—bumetanide solution (0.1 mmol/L), Ctr—control conditions (trachea samples incubated in RS, A and B solution); Smoke—smoking conditions (trachea samples exposed to cigarette smoke and incubated in RS, A, and B solution).

## 4. Discussion

The measurement of electrophysiological parameters enables the evaluation of ion transport mechanisms occurring in the organ and tissue [8–11,13,14,18,24,25]. Measurements of the resistance and the transepithelial electric potential during stimulation and in

stationary conditions (i.e., without stimulation) depend on changes in the transport of ions occurring in the cells that build the organ [9,14,18,24]. In the presented experiment, the airway fragments (tracheas from experimental animals) were used as the research model, because the respiratory tract is most exposed to cigarette smoke and other pollutants and it is the first to come into contact with smoke. The use of laboratory animals for research on the pathophysiology of changes in ion transport is justified due to the similarity in the production of ASL and the distribution of channels and ion transporters in the respiratory epithelium [26]. The examined tissue fragments were full-thickness, with preserved nerve endings and physiological possibilities of changes in the operation of channels, transporters, and pumps involved in the transport of ions and water [9,14,18,24].

Measurement of the PD in stationary conditions, i.e., without stimulation, reflects the constantly occurring transport of ions [9,14,18,24]. Healthy tissue has the ability to change the ion transport pathway in order to maintain a negative charge on the surface of the airways and maintain the ability to react to stimuli, produce ASL and the cough reflex, by opening or closing selected transporters/channels, or intensifying the work of ion pumps [9,14,24–28]. The applied model of incubation of specimens in the solution of amiloride, a sodium ion transport blocker, forces an increased transport of chloride ions, while incubation in the solution of bumetanide, a chloride ion transport blocker, intensifies the transport of sodium ions in the examined tissue [9,12,14,18,24]. During the experiments, for both the controls and the samples exposed to cigarette smoke, PD measurements were made during 30 min of each experiment. For the control fragments incubated in isosmotic RS and B, there were no changes in PD during 30 min of measurements. There was a significant increase in PD for the samples incubated in A, both for the controls and for trachea samples exposed to cigarette smoke (Table 4, Wilcoxon test). It seems that the use of a sodium ion transport inhibitor causes a less intensive transport of chloride ions and cigarette smoke does not change this mechanism. This is consistent with data showing that short-term exposure to cigarette smoke inhibits cystic fibrosis transmembrane conductance regulator (CFTR) function and transepithelial chloride ion transport [25,27–30]. CFTR is an ATP-gated chloride channel found in the apical membranes of epithelial cells lining surfaces of several organs, including airways [31]. Dysregulation of CFTR function is associated, among other things, with the activity of the mitogen-activated protein (MAP) and c-JUN n-terminal kinases and the p38 protein, which may initiate processes leading to carcinogenesis [25,32]. Under physiological conditions, maintaining a negative charge on the surface of the respiratory tract depends primarily on the activity of the sodium-potassium pump, the secretion of chloride ions, and the absorption of sodium ions in equal proportions [9,12,14,18,24]. In the experimental model used, a 30-min exposure to cigarette smoke inhibited the sodium-potassium pump and altered the ongoing ion transport, as PD was significantly elevated compared to controls under all incubation conditions (Table 5, Mann-Whitney test). The diminished sodium-potassium pump activity may be related to the release of calcium on the surface of the respiratory tract under the influence of cigarette smoke, which causes a calcium deficiency for basic cellular processes [25,28]. In addition, the lack of calcium and ATP causes the inability to regulate ion channels, affects the production of less hydrated ASL, and slows down the action of cilia [25,27,28]. Inhibited ion transport affects the dehydration of ASL in the respiratory tract and causes inhibition of cilia movement [22,25]. Reducing the amount of water in ASL leads to drying of the mucous membranes and increased diffusion of substances from smoke into the body [33]. Moreover, cigarette smoke causes an increase in the secretion of transforming growth factor (TGF) from the cells of the respiratory epithelium, which, together with a decrease in the amount of calcium, additionally increases the permeability of the epithelium [22,25] and further airway dysfunction [23]. Interestingly, Astrand et al. [19] proved that blocking ENaC in the respiratory tract can be an effective tool for restoring proper hydration of ASL and can stimulate cilia to evacuate accumulated mucus. Thus, the use of amiloride for rinsing of the airways may result in the restoration of the physiological functions of ASL which have been dried up after exposure to cigarette smoke.

The application of a mechanical stimulus (isoosmotic RS) or a mechanical-chemical stimulus (solutions A and B) in the form of a gentle flow of fluid over the surface of the epithelium caused measurable and reproducible changes in the transport of ions in the examined airway fragments. Ion transport occurring during stimulation was always significantly different from that occurring continuously, i.e., in stationary conditions, for all experimental groups. After the end of the stimulation, the potential returned to the initial level. It was shown that cigarette smoke exposure did not modify the tissue response to the stimulus, but the responses of the smoke-exposed trachea specimens were significantly different from the controls. Tissue samples exposed to cigarette smoke and incubated in RS reacted more intensely to the mechanical stimulus. The intensity of the reaction resulted from the spontaneous transport of sodium and chloride ions, the samples reacted with the increased absorption of sodium ions and/or secretion of chloride ions (Table 3). The trachea overreacted during the reception of stimuli; the transport of ions was more intense. The increase in the intensity of ion transport is associated with a change in water flow and gradual depletion of the cells. The presented results prove that even a single 30-min contact with cigarette smoke initiates mechanisms that might participate in the development of a cough reflex, hypersensitivity reactions, and activate the defense processes of respiratory epithelial cells. Excessive ion transport and depletion of cellular calcium and ATP resources after exposure to smoke may contribute to the increase in the activity of MAP and JUN kinases, which may trigger pathways leading to cell division and initiation of carcinogenesis [22]. However, the use of ion transport inhibitors, both A and B, resulted in a decrease in the measured PDmin. Blocking the transport pathway of one of the ions after exposure to cigarette smoke reduced the ability to regulate the transport of the other ion and diminish the clearance process, whereas tissue samples not exposed to smoke could clean themselves [19,26]. Therefore, cigarette smoke reduced the ability to react to stimuli related to ion transport mechanisms, i.e., the cough reflex, ASL hydration, and the release of immunologically active substances [33].

It has been proven that short-term exposure to cigarette smoke intensifies the transport of sodium ions by opening ENaC and stimulating the transient receptor potential cation channel subfamily V, member 1 (TRPV1) [34]. The activation of TRPV1 is correlated with the stimulation of immunocompetent cells and the initiation of an inflammatory reaction. In addition, the increase in $Na^+$ absorption causes the overproduction of mucus, the task of which is to remove impurities from cigarette smoke, but it also stimulates immunocompetent cells to secrete interleukin 1 (IL-1), which sustains inflammation and initiates airway remodeling [35].

During the experiment, all tracheal fragments remained viable and fully reactive [9,14,18,24]. There was no significant decrease in electrical resistance, which would indicate a loss of tissue compactness or viability. Under incubation conditions in RS, the R measurement showed an increasing trend for the control fragments. Tissue samples that were not exposed to cigarette smoke had increased permeability to ions, which can be considered as a defensive reaction [27]. The respiratory tract reacts to changing conditions, i.e., the administration of ion transport inhibitors A and B in the stimulation solution during the experiment. The incubation of both the controls and smoke experimental group in every type of solution (RS, A, and/or B) did not cause changes in the R values during the experiment (Table 4). However, R was significantly decreased in the smoke-exposed samples incubated in RS and B in comparison to the control samples. Thus, the tissue lost its ability to rapidly modify ion and water transport under the influence of changing conditions [27]. As a result, there may be a risk of drying the airway surface. In addition, after contact with cigarette smoke and incubation in RS and B, the measured R values significantly decreased, which may be related to the failure of a rapid change in the transport of sodium ions through the entire thickness of the airways [27]. The mucus accumulating on the surface may then be thickened and more difficult to evacuate [9,14,18,24].

## 5. Conclusions

In the presented study, it was proven that a 30-min exposure to cigarette smoke caused significant changes in the transport of sodium and chloride ions occurring constantly in the airways and during their stimulation with a mechanical stimulus. Reactions to the stimulations were greater than those of the trachea samples not exposed to smoke, which may be associated with the initiation of the cough reflex and hypersensitivity reactions. The results of the presented study indicate that the airways exposed to cigarette smoke became more permeable to ions, water, and substances with which they came into contact. Since exposure to chemicals that are emitted during smoking has been shown to have irreversible and harmful effects on the body, it seems to be extremely important that even a single short exposure to cigarette smoke is associated with increased permeability of the tracheal epithelium, which might be sufficient to create the possibility of initiating procarcinogenic processes.

**Author Contributions:** K.H.—conceptualization, data curation, investigation, visualization, software, writing—original draft preparation; I.B.—data curation, investigation, writing—review and editing; E.C.—investigation, writing—review and editing; A.B.—investigation, writing—review and editing; E.P.—data curation, writing—review and editing; D.O.-S.—formal analysis, writing—review and editing; A.W.—project administration, writing—review and editing; K.S.-G.—formal analysis, supervision, validation, writing—review and editing; I.H.-I.—conceptualization, investigation, methodology, writing—original draft preparation. All authors have read and agreed to the published version of the manuscript.

**Funding:** This research received no external funding.

**Institutional Review Board Statement:** The present experiment did not include living animals and according to the Polish and European Union law, the bioethical committee agreement was not required. Animal care was in accordance with the guidelines and regulations as stipulated by the Polish Animal Protection Act and the European Directive on the Protection of Animals Used for Scientific Purposes (2010/63/EU). All applicable institutional and national guidelines for the care and use of animals were followed.

**Informed Consent Statement:** Not applicable.

**Data Availability Statement:** Raw data supporting the findings of this study are available from the corresponding author (I.H.I.) on request.

**Conflicts of Interest:** The authors declare no conflict of interest.

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
