# Peer review of "30-Min Exposure to Tobacco Smoke Influences Airway Ion Transport—An In Vitro Study"

_curroncol, doi:10.3390/curroncol30070508_

Round 1

Reviewer 1 Report

This article estimates how a 30-minute exposure period to tobacco smoke may influence the airway ion as tobacco smoke inhalation injury is one of the predominant hazards to the respiratory epithelium. The introduction is clear and concise and gives a good rationale for developing a tracheal model for rabbits. All the figures and tables are easily readable, correct and informative. This paper presents novelty and originality of statistic research, importance and impact of results. Although, the 30 minute exposure is as very short period of time; findings summarized a strong argument.

Please, the authors may comment:

-  measured any inflammatory parameters as might alter the detected values

-any differences among quartile of trachea between male/female rabbits

Author Response

We would like to thank the Reviewer for the careful and insightful review of our manuscript and for the positive evaluation of our article.

  1. Measured any inflammatory parameters as might alter the detected values
  2. 1 According to the Reviewer recommendation, we have added to the discussion the information concerning the influence of cigarette smoke on the development of inflammation in conjunction with the changes in ion transport (page 10, lines 1-8).

  1. Any differences among quartile of trachea between male/female rabbits
  2. 2 In our research, we try to be guided by ethical considerations and only kill as many animals as is absolutely necessary. For this reason, the measured parameters of the control group, namely PD, PDmin and R, were developed on the basis of previously collected data from the conducted experiments. In these studies, differences in the studied parameters depending on the sex of the animals were not determined, therefore we are not able to provide such a comparison without performing new experiments and sacrificing new animals, which is not possible at the moment. However, we thank the Reviewer for pointing out this gap in the research so far, in future experiments we will look at possible differences between males and females.​

Reviewer 2 Report

Article review

«30-minute exposure to tobacco smoke influences airway ion 2 transport – an in vitro study»

The article presents the results of a study of the short-term effects of cigarette smoke on rabbit tracheal cells. This study is relevant, since most of the previous studies are based on long-term (for several weeks in the experiment, decades - in the epidemiological studies) inhalation of tobacco smoke. And many public awareness campaigns talk about the increased risks of lung cancer, chronic obstructive pulmonary disease, osteoporosis, early menopause, and other diseases that are common in people over 50 years old. Knowledge of the early pathological changes that occur in the body even with a short duration of tobacco consumption will help to further argue the need to quit smoking at a young age.

The article is divided into sections: summary, introduction, materials and methods, results, discussion, conclusion.

The research methods are described in detail with reference to previous studies on the applied model; statistical data processing was carried out in the Statistica 13.1 program.

The discussion of their own results with the involvement of data from other studies was carried out in a very interesting and qualified manner. A possible explanation of the pathophysiological mechanisms of the changes is given. For example like this one: “The diminished sodium-potassium pump activity may be related to the release of calcium on the surface of the respiratory tract under the influence of cigarette smoke, which causes a calcium deficiency for basic cellular processes”.

However, there are some comments and questions to be cleared for an improvement of the article.

1.

The summary is not structured and the main results of the study are not presented. It is recommended to highlight sections: "Materials and Methods", "Results", "Conclusion". In the "Results" section, it is necessary to provide the main data obtained with figures to justify the conclusions.

2.

The introduction presents the results of epidemiological studies and reviews that indicate the development of various diseases in adults and children associated with both tobacco smoking and inhalation of environmental tobacco smoke (passive smoking). A table of carcinogenic and toxic substances that can cause carcinogenesis is given. This part of the introduction is not directly related to the subject of the article, it is a listing of known facts without specifying the specific risks of developing a particular disease, which, in our opinion, is redundant, and this part of the introduction should be shortened or deleted.

3.

The results of the research are presented in four tables, but these results are difficult for the reader to understand: the presentation of the data is not illustrative. It is desirable to divide tables 2 and 3 into a table of the actual measurement data with a presentation of the significance of differences between groups, and present the changes in the results on a graph depending on the modes: RS, A and B.

4.

The discussion section describes the Ussing chamber, which is a description of the research method, so this part of the discussion should be moved to the materials and methods section.

5.

In the conclusions you state that “the presented results prove that even a single, 30-minute contact with cigarette smoke causes a cough reflex, hypersensitivity reaction and activates the defense processes of respiratory epithelial cells”;  “it seems to be extremely important that a single exposure to cigarette smoke is enough to create the possibility of initiating a series of biochemical reactions leading to carcinogenesis”. What are the grounds for these statements? This was not studied in the work.

6.

Please describe the actuality of the presented work in comparison with the works in the references 19, 26? (Blocking the transport pathway of one of the ions after exposure to cigarette smoke reduced the ability to regulate the transport of the other ion and diminish clearance process, whereas tissue samples not exposed to smoke could clean themselves [19,26]). 

Author Response

Answer Reviewer 2#

We would like to thank the Reviewer for the careful and insightful review of our manuscript and for the positive evaluation of our article.

  1. The summary is not structured and the main results of the study are not presented. It is recommended to highlight sections: "Materials and Methods", "Results", "Conclusion". In the "Results" section, it is necessary to provide the main data obtained with figures to justify the conclusions.
  2. 2 The summary has been corrected according to the Reviewer’s suggestion.

  1. The introduction presents the results of epidemiological studies and reviews that indicate the development of various diseases in adults and children associated with both tobacco smoking and inhalation of environmental tobacco smoke (passive smoking). A table of carcinogenic and toxic substances that can cause carcinogenesis is given. This part of the introduction is not directly related to the subject of the article, it is a listing of known facts without specifying the specific risks of developing a particular disease, which, in our opinion, is redundant, and this part of the introduction should be shortened or deleted.
  2. 2 Thank you for the recommendation. The introducton section has been shortened.

  1. The results of the research are presented in four tables, but these results are difficult for the reader to understand: the presentation of the data is not illustrative. It is desirable to divide tables 2 and 3 into a table of the actual measurement data with a presentation of the significance of differences between groups, and present the changes in the results on a graph depending on the modes: RS, A and B.
  2. 3 We would like to point out that the tables 2 and 3 contain only the actual measurement data, while the significance of differences between groups is presented in the tables 4 and 5. We believe that such a presentation is the most readable due to the complicated study scheme. This method of presentation was used by our team in many previous articles on research using an analogous methodology (testing of tissue electrophysiological parameters). However, we want to thank the Reviewer for this suggestion. In the future studies and the preparation of the next article, we will try to present the results using graphs.

  1. The discussion section describes the Ussing chamber, which is a description of the research method, so this part of the discussion should be moved to the materials and methods section.
  2. 4 We have moved the description of the Ussing chamber from the discussion section to the material and methodology section.

  1. In the conclusions you state that “the presented results prove that even a single, 30-minute contact with cigarette smoke causes a cough reflex, hypersensitivity reaction and activates the defense processes of respiratory epithelial cells”;  “it seems to be extremely important that a single exposure to cigarette smoke is enough to create the possibility of initiating a series of biochemical reactions leading to carcinogenesis”. What are the grounds for these statements? This was not studied in the work.
  2. 5 The conclusions have been corrected according to the Reviewer’s recommendation.

  1. Please describe the actuality of the presented work in comparison with the works in the references 19, 26? (Blocking the transport pathway of one of the ions after exposure to cigarette smoke reduced the ability to regulate the transport of the other ion and diminish clearance process, whereas tissue samples not exposed to smoke could clean themselves [19,26]). 
  2. 6 In the discussion section, the studies published by Åstrand et al. [19] (page 9, lines 21-25) and Hahn at al. [26] have been discussed in more detail (page 8, lines 20-23).

Round 2

Reviewer 1 Report

In my point of view, the paper can be published after the revision form